

# Multipole theory of optical spatial dispersion in crystals

Óscar Pozo Ocaña[1] and Ivo Souza[1,2]

**1** Centro de Física de Materiales, Universidad del País Vasco, 20018 San Sebastián, Spain
**2** Ikerbasque Foundation, 48013 Bilbao, Spain

## Abstract

Natural optical activity is the paradigmatic example of an effect originating in the weak spatial inhomogeneity of the electromagnetic field on the atomic scale. In molecules, such effects are well described by the multipole theory of electromagnetism, where the coupling to light is treated semiclassically beyond the electric-dipole approximation. That theory has two shortcomings: it is limited to bounded systems, and its building blocks – the multipole transition moments – are origin dependent. In this work, we recast the multipole theory in a translationally-invariant form that remains valid for crystals. Working in the independent-particle approximation, we introduce "intrinsic" multipole transition moments that are origin independent and transform covariantly under gauge transformations of the Bloch eigenstates. Electric-dipole transitions are given by the interband Berry connection, while magnetic-dipole and electric-quadrupole transitions are described by matrix generalizations of the intrinsic magnetic moment and quantum metric. In addition to multipole-like terms, the response of crystals at first order in the wavevector of light contains band-dispersion terms that have no counterpart in molecular theories. The full response is broken down into magnetoelectric and quadrupolar parts, which can be isolated in the static limit where electric and magnetic fields become decoupled. The rotatory-strength sum rule for crystals is found to be equivalent to the topological constraint for a vanishing chiral magnetic effect in equilibrium, and the formalism is validated by numerical tight-binding calculations.

doi:[10.21468/SciPostPhys.14.5.118]()

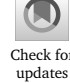
# 1 Introduction

As the wavelength of optical radiation is large compared to atomic dimensions, the interaction of light with matter is generally well described by taking the long-wavelength limit (electric-dipole approximation). In that approximation, the response of the medium to an electromagnetic perturbation is treated as local in space. When nonlocality is taken into account, the response acquires a dependence on the wavevector $\mathbf{q}$ of light, and this is known as spatial dispersion [1, 2].

Although the effects of spatial dispersion can often be treated as small corrections, they are significant in that they lead to qualitatively new phenomena. One example is natural optical activity [1], whose most familiar manifestation is the rotation of the plane of polarization of linearly-polarized light travelling through chiral molecules in solution. Lesser-known manifestations of spatial dispersion include gyrotropic birefringence and nonreciprocal directional dichroism [3, 4]; these are magneto-optical effects that occur in acentric magnetic materials without requiring a net magnetization.

Because of the fundamental and industrial importance of chiral molecules, molecular quantum theories of natural optical activity – and, by extension, of other spatially-dispersive optical effects – have been developed over many decades, on the basis of the multipole theory of electromagnetism [5–8]. This has led to the development, starting in the mid 1990s, of several *ab initio* methods for calculating optical rotatory dispersion and natural circular dichroism spectra of molecules. Some of those methods rely on sum-over-states formulas [9]; in others, the explicit summation over a truncated set of excited states is avoided using either static-limit [10] or finite-frequency [11] linear-response schemes, or real-time propagation approaches [11,12] (see Refs. [13–15] for reviews).

By comparison, there have been relatively few attempts to formulate bulk theories of optical spatial dispersion [16–18], particularly within the one-electron band picture [19–26]. As a result, only a small number of *ab initio* calculations of natural optical activity [20,21,27–29] and of nonreciprocal directional dichroism [30,31] have been carried out for crystals. Such effects provide valuable information about broken structural and magnetic symmetries, and their study in novel bulk [30–32] and quasi-two-dimensional [33–37] materials calls for improved theoretical descriptions.

In this work, we develop a microscopic theory of optical spatial dispersion in crystals that is firmly rooted in the molecular multipole theory. We work in the independent-particle approximation neglecting local-field effects [21], and focus on the electronic response with frozen ions. We proceed by evaluating the optical conductivity at first order in $\mathbf{q}$, including both interband and intraband contributions, and arrive at a sum-over-states expression in terms of well-defined multipole-like transition moments.

To set the stage, let us introduce the multipole transition moments as defined in the standard molecular theory [7,8]. The electric dipole (E1) appears at leading order in the multipole expansion, followed at the next order by the magnetic dipole (M1) and electric quadrupole (E2). These are the needed ingredients to describe natural optical activity, gyrotropic birefringence, and nonreciprocal directional dichroism. In the independent-particle approximation, they take the form

$$\mathbf{d}_{nl} = -e\langle\phi_n|\mathbf{r}|\phi_l\rangle, \tag{1a}$$

$$\mathbf{m}_{nl} = -\frac{e}{2}\langle\phi_n|\mathbf{r}\times\mathbf{v}|\phi_l\rangle, \tag{1b}$$

$$q_{nl}^{ab} = -e\langle\phi_n|r_a r_b|\phi_l\rangle, \tag{1c}$$

where $\phi_n(\mathbf{r})$ and $\phi_l(\mathbf{r})$ are occupied and empty energy eigenstates of the molecule, respectively, and $-e$ is the electron charge. (The M1 transition moment also has a spin part; we omit it for now, but it will be included later.)

In trying to extend the multipole theory to periodic crystals, one is faced with the problem of how to define the transition moments when the molecular orbitals $\phi_n(\mathbf{r})$ are replaced by Bloch eigenstates $\psi_{n\mathbf{k}}(\mathbf{r}) = e^{i\mathbf{k}\cdot\mathbf{r}}u_{n\mathbf{k}}(\mathbf{r})$, given that the matrix elements in Eq. (1) involve the nonperiodic position operator $\mathbf{r}$. For E1 transitions between nondegenerate bands $n$ and $l$, there is a well-known prescription, namely $\mathbf{d}_{nl}(\mathbf{k}) = -e\langle u_{n\mathbf{k}}|i\partial_{\mathbf{k}}u_{l\mathbf{k}}\rangle$ [38,39].

The situation is less clear when it comes to defining M1 and E2 transition moments in the Bloch representation. Already for molecules, their definitions in Eq. (1) are somewhat problematic, as they give values that change under a rigid shift of the coordinate system. Molecular properties should be origin independent, and for spatially-dispersive optical coefficients that is generally ensured by a cancellation between the origin dependences of different terms of the same order in the multipole expansion [6–9]. This is not entirely satisfactory from a formal standpoint, and moreover it leads to slightly origin-dependent numerical results, because the cancellation is not exact for incomplete basis sets [8,9].

In our independent-particle formulation, the optical conductivity at first order in $\mathbf{q}$ is written in terms of "intrinsic" multipole transition moments $\bar{\text{E}}1$, $\bar{\text{M}}1$, and $\bar{\text{E}}2$ that are origin independent and well defined for both molecules and crystals. For molecules, these modified transition moments take the form

$$\bar{\mathbf{d}}_{nl} = -e\langle\phi_n|\mathbf{r}-(\bar{\mathbf{r}}_n+\bar{\mathbf{r}}_l)/2|\phi_l\rangle, \tag{2a}$$

$$\bar{\mathbf{m}}_{nl} = -\frac{e}{2}\langle\phi_n|[\mathbf{r}-(\bar{\mathbf{r}}_n+\bar{\mathbf{r}}_l)/2]\times\mathbf{v}|\phi_l\rangle, \tag{2b}$$

$$\bar{q}_{nl}^{ab} = -e\langle\phi_n|[r_a-(\bar{r}_n^a+\bar{r}_l^a)/2][r_b-(\bar{r}_n^b+\bar{r}_l^b)/2]|\phi_l\rangle. \tag{2c}$$

As they are defined relative to an intrinsic origin located halfway between the centers $\bar{\mathbf{r}}_n = \langle\phi_n|\mathbf{r}|\phi_n\rangle$ and $\bar{\mathbf{r}}_l = \langle\phi_l|\mathbf{r}|\phi_l\rangle$ of the two orbitals, the matrices $\bar{d}$, $\bar{m}$, and $\bar{q}$ are manifestly origin independent. For crystals, we find that they acquire "quantum-geometric" forms when expressed in terms of the cell-periodic Bloch eigenstates: $\bar{d}$ is given by the interband Berry connection, while $\bar{m}$ and $\bar{q}$ are described by generalizations – with both intraband and interband parts – of the intrinsic orbital moment [40] and of the quantum metric [41], respectively.

The intraband orbital moment and quantum metric were already known to contribute to the spatially-dispersive optical response in metals [23–25]; we clarify that the quantum metric appears quite generally and not just in two-band models, and identify additional Fermi-surface terms. As for the interband counterparts of the intrinsic orbital moment and quantum metric, they had not been clearly identified in previous theoretical studies of spatial dispersion in band insulators [19–22], where the optical matrix elements were written in velocity form, and without isolating the magnetic and quadrupolar parts.

The paper is organized as follows. In Sec. 2, we introduce some basic definitions and relations. Section 3 contains the derivation and analysis of our main result: an expression for the bulk optical conductivity at first order in **q**. The derivation is split into several steps. We start in Sec. 3.1 from the Kubo formula for the frequency- and wavevector-dependent optical conductivity in the velocity gauge. That formula suffers from apparent divergences at zero frequency, and we recast it in a form that is manifestly divergence free. We then expand the Kubo formula to linear order in **q**, treating the **q** dependence coming from the band dispersion and from the matrix elements in Secs. 3.2 and 3.3, respectively. In Sec. 3.4, we convert the optical matrix elements from velocity to length form, and express them in terms of intrinsic multipole transition moments between Bloch eigenstates. In Sec. 3.5 we collect terms, and report the final expression for the optical conductivity at first order in **q**. Finally, in Sec. 3.6 we briefly discuss its decomposition into magnetoelectric and quadrupolar parts, and the associated physical effects in the static limit. In Sec. 4 we consider the molecular limit of our formalism, and its relation to the standard multipole theory. The formal part of the paper ends in Sec. 5 with an analysis of optical sum rules, and in Sec. 6 we present numerical results for a tight-binding model. We conclude in Sec. 7 with a summary and discussion.

## 2 Basic definitions and relations

Consider the response of a medium to a monochromatic electromagnetic field. We work in the "temporal gauge" [2], where the electromagnetic field is fully described by the vector potential $\mathbf{A}(t,\mathbf{r}) = \text{Re}\left[\mathbf{A}(\omega,\mathbf{q})e^{i(\mathbf{q}\cdot\mathbf{r}-\omega t)}\right]$. To linear order in the field amplitude, the induced current density reads

$$j_a(\omega,\mathbf{q}) = \Pi_{ab}(\omega,\mathbf{q})A_b(\omega,\mathbf{q}) = \frac{1}{i\omega}\Pi_{ab}(\omega,\mathbf{q})E_b(\omega,\mathbf{q}), \tag{3}$$

so that one may define an effective conductivity as $\sigma_{ab}(\omega,\mathbf{q}) = (1/i\omega)\Pi_{ab}(\omega,\mathbf{q})$ [1–3]. If the spatial dispersion is weak, the effective conductivity can be expanded as

$$\sigma_{ab}(\omega,\mathbf{q}) = \sigma_{ab}(\omega,\mathbf{0}) + \sigma_{ab,c}(\omega)q_c + \mathcal{O}(q^2), \tag{4}$$

where a summation over the repeated Cartesian index $c$ is implied. The zeroth-order term is the optical conductivity in the long-wavelength limit, that is, in the electric-dipole approximation. The next term in the expansion captures the effects of spatial dispersion to the order of magnetic dipoles and electric quadrupoles, and will be the focus of our study. Let us split it into symmetric (S) vs antisymmetric (A), and into Hermitian (H) vs anti-Hermitian (AH) parts with respect to its first two indices,

$$\sigma_{ab,c}^{\text{S}} = \text{Re}\,\sigma_{ab,c}^{\text{H}} + i\,\text{Im}\,\sigma_{ab,c}^{\text{AH}}, \tag{5a}$$

$$\sigma_{ab,c}^{\text{A}} = \text{Re}\,\sigma_{ab,c}^{\text{AH}} + i\,\text{Im}\,\sigma_{ab,c}^{\text{H}}. \tag{5b}$$

The H and AH parts are absorptive and reactive, respectively [1, 2]. Regarding the S and A parts, they transform differently when time reversal $\mathcal{T}$ is applied to the material, that is, under

reversal of its magnetic order parameter. According to the Onsager reciprocity relation [1,2], the A part is $\mathcal{T}$ even, and the S part is $\mathcal{T}$ odd; the former describes natural optical activity, and the latter describes spatially-dispersive magneto-optical effects. The entire $\sigma_{ab,c}$ tensor is odd under spatial inversion $\mathcal{P}$, and hence it vanishes in centrosymmetric systems. If both $\mathcal{P}$ and $\mathcal{T}$ are broken but the combined $\mathcal{PT}$ symmetry is present, $\sigma_{ab,c}^{\mathrm{A}}$ vanishes but $\sigma_{ab,c}^{\mathrm{S}}$ can be nonzero.

# 3 The bulk formula for $\sigma_{ab,c}(\omega)$

## 3.1 Kubo formula for the effective conductivity $\sigma_{ab}(\omega, \mathbf{q})$

We now specialize to a three-dimensional crystal described by a single-particle Pauli Hamiltonian $\mathcal{H}$ [38] with a local external potential, and introduce the electromagnetic perturbation via an interaction Hamiltonian $\mathcal{H}_{\mathrm{I}}$ expressed in the velocity gauge [42]. To linear order in the vector potential $\mathbf{A}(t, \mathbf{r})$, the interaction Hamiltonian can be written as [23]

$$\mathcal{H}_{\mathrm{I}}^{(\eta)}(t, \mathbf{r}) = \frac{e}{2}\left[\tilde{\mathbf{A}}^{(\eta)}(t, \mathbf{r}) \cdot \mathbf{v} + \mathbf{v} \cdot \tilde{\mathbf{A}}^{(\eta)}(t, \mathbf{r})\right] + \frac{e}{m_e}\mathbf{S} \cdot \partial_{\mathbf{r}} \times \tilde{\mathbf{A}}^{(\eta)}(t, \mathbf{r}), \tag{6}$$

where $m_e$ is the electron mass, $\mathbf{v} = (1/i\hbar)[\mathcal{H}, \mathbf{r}]$ is the unperturbed velocity operator, and $\mathbf{S}$ is the spin operator. In addition, we have defined $\tilde{\mathbf{A}}^{(\eta)}(t, \mathbf{r}) = e^{\eta t}\mathbf{A}(t, \mathbf{r})$, where the parameter $\eta$ is formally a positive infinitesimal that controls the adiabatic turning on of the coupling between the electromagnetic field and the crystal [8,43–46].

A standard perturbative calculation yields the following Kubo formula for the optical conductivity in the spectral representation [23,44],

$$\sigma_{ab}^{(\eta)}(\omega, \mathbf{q}) = \delta_{ab}\frac{ie^2 N}{m_e \omega} + \frac{ie^2}{\hbar\omega}\sum_{n,l}\int_{\mathbf{k}}\frac{f_{ln\mathbf{k}}(\mathbf{q})}{\omega_{ln\mathbf{k}}(\mathbf{q}) - \omega - i\eta}\mathcal{M}_{nl\mathbf{k}}^{ab}(\mathbf{q}). \tag{7}$$

Here $\int_{\mathbf{k}} = \int_{\mathrm{BZ}} d\mathbf{k}/(2\pi)^3$, $\omega_{ln\mathbf{k}}(\mathbf{q}) = \omega_l(\mathbf{k} + \mathbf{q}/2) - \omega_n(\mathbf{k} - \mathbf{q}/2)$, where $\hbar\omega_n(\mathbf{k}) = \varepsilon_n(\mathbf{k})$ is the band energy, and $f_{ln\mathbf{k}}(\mathbf{q}) = f_l(\mathbf{k} + \mathbf{q}/2) - f_n(\mathbf{k} - \mathbf{q}/2)$, where $f_n(\mathbf{k}) = f[\omega_n(\mathbf{k})]$ is the Fermi-Dirac occupation factor. In the first term, $N$ is the total number of electrons per unit volume; in the second, the matrix element is defined as

$$\mathcal{M}_{nl\mathbf{k}}^{ab}(\mathbf{q}) = \left[I_{ln\mathbf{k}}^a(\mathbf{q})\right]^* I_{ln\mathbf{k}}^b(\mathbf{q}), \tag{8}$$

where $\mathbf{I}_{ln\mathbf{k}}(\mathbf{q})$ is a sum of orbital and spin contributions [23],

$$\mathbf{I}_{ln\mathbf{k}}^{\mathrm{orb}}(\mathbf{q}) = \langle u_l(\mathbf{k} + \mathbf{q}/2)|\mathbf{v}(\mathbf{k})|u_n(\mathbf{k} - \mathbf{q}/2)\rangle, \tag{9a}$$

$$\mathbf{I}_{ln\mathbf{k}}^{\mathrm{spin}}(\mathbf{q}) = \frac{ig_s}{2m_e}\langle u_l(\mathbf{k} + \mathbf{q}/2)|\mathbf{S}|u_n(\mathbf{k} - \mathbf{q}/2)\rangle \times \mathbf{q}. \tag{9b}$$

In Eq. (9a), $\mathbf{v}(\mathbf{k}) = (1/\hbar)\partial_{\mathbf{k}}H(\mathbf{k})$ with $H(\mathbf{k}) = e^{-i\mathbf{k}\cdot\mathbf{r}}\mathcal{H}e^{i\mathbf{k}\cdot\mathbf{r}}$, and in Eq. (9b), $g_s \approx 2$ is the spin $g$-factor of the electron. Henceforth, the index $\mathbf{k}$ will be omitted for brevity.

The $1/\omega$ prefactors in Eq. (7), inherited from Eq. (3), make it singular at $\omega = 0$. That singularity is only apparent [46], and it can be removed as follows. First, split Eq. (7) into reactive and absorptive parts using $\lim_{\eta\to 0^+}(x - i\eta)^{-1} = 1/x + i\pi\delta(x)$. Next, notice that the reactive part can be rewritten by invoking the Kramers-Krönig relation

$$\sigma_{ab}^{\mathrm{AH}}(\omega_0, \mathbf{q}) = -\frac{i}{\pi}\mathrm{P}\int_{-\infty}^{\infty} d\omega\,\frac{\sigma_{ab}^{\mathrm{H}}(\omega, \mathbf{q})}{\omega - \omega_0}, \tag{10}$$

while in the absorptive part the factor $1/\omega$ can be replaced with $1/\omega_{ln}$ thanks to the delta function. Finally, recombine the two parts to obtain

$$\sigma_{ab}(\tilde{\omega}, \mathbf{q}) = \frac{e^2}{\hbar} \sum_{n,l} \int_{\mathbf{k}} \mathcal{E}_{ln}(\tilde{\omega}, \mathbf{q}) \mathcal{M}_{nl}^{ab}(\mathbf{q}), \tag{11}$$

where we have defined $\tilde{\omega} = \omega + i\eta$ and

$$\mathcal{E}_{ln}(\tilde{\omega}, \mathbf{q}) = \frac{f_{ln}(\mathbf{q})}{\omega_{ln}(\mathbf{q})} \frac{i}{\omega_{ln}(\mathbf{q}) - \tilde{\omega}}. \tag{12}$$

At zeroth order in $\mathbf{q}$ Eq. (11) reduces to Eq. (25) of Ref. [46], and at first order it becomes, in the notation of Eq. (4),

$$\sigma_{ab,c}(\tilde{\omega}) = \frac{e^2}{\hbar} \sum_{n,l} \int_{\mathbf{k}} \left[ \mathcal{E}_{ln}^{;c}(\tilde{\omega}) \mathcal{M}_{nl}^{ab}(\mathbf{0}) + \mathcal{E}_{ln}(\tilde{\omega}, \mathbf{0}) \mathcal{M}_{nl}^{ab,c} \right]. \tag{13}$$

In Sec. 5, we discuss how the equivalence between the Kubo formulas (7) and (11) at zeroth and first orders in $\mathbf{q}$ is related to the oscillator-strength and rotatory-strength sum rules, respectively. In the context of tight-binding calculations the diamagnetic term in Eq. (7) changes form [47], while Eq. (11) remains unchanged.

In the following subsections, we evaluate the expansion coefficients of the $\mathcal{E}$ and $\mathcal{M}$ matrices appearing in Eq. (13). We start in Sec. 3.2 with the expansion of $\mathcal{E}$, and then devote Secs. 3.3 and 3.4 to the expansion and subsequent manipulations of $\mathcal{M}$, which is where our treatment differs more substantially from that of previous works.

The terms in the resulting expression for $\sigma_{ab,c}(\tilde{\omega})$ can be classified as either "band dispersive" or "molecular", depending on whether or not they vanish for a crystal composed of nonoverlapping units. The first term in Eq. (13) is clearly band dispersive, because the quantity $\mathcal{E}_{ln}^{;c}(\tilde{\omega})$ involves the band velocities $\mathbf{v}_n = \partial_{\mathbf{k}} \omega_n$ (see Eqs. (14c) and (14d) below). While less obvious, the second term in Eq. (13) is not purely molecular; as we will see, it has a band-dispersion component that went unnoticed in previous works [19, 22].

## 3.2 Expansion in q of the band-energy terms

When expanding Eq. (12) in powers of $\mathbf{q}$, the intraband ($l = n$) and interband ($l \neq n$) parts must be treated separately. To first order in $\mathbf{q}$, one finds

$$\mathcal{E}_{nn}(\tilde{\omega}, \mathbf{0}) = -\frac{i}{\tilde{\omega}} f_n', \tag{14a}$$

$$\mathcal{E}_{ln}(\tilde{\omega}, \mathbf{0}) = i \frac{f_{ln}}{\omega_{ln}} (\omega_{ln} + \tilde{\omega}) Z_{ln}(\tilde{\omega}), \tag{14b}$$

$$\mathcal{E}_{nn}^{;c}(\tilde{\omega}) = -\frac{i}{\tilde{\omega}^2} f_n' v_n^c, \tag{14c}$$

$$\mathcal{E}_{ln}^{;c}(\tilde{\omega}) = \frac{i}{2} \frac{v_l^c f_l' + v_n^c f_n'}{\omega_{ln}} (\omega_{ln} + \tilde{\omega}) Z_{ln}(\tilde{\omega})$$
$$- i f_{ln} \frac{Z_{ln}^2(\tilde{\omega})}{\omega_{ln}^2} \left[ (\omega_{ln} + \tilde{\omega})^2 (\omega_{ln} - \tilde{\omega}/2) \right] (v_l^c + v_n^c), \tag{14d}$$

where $f_n' = \partial f_n / \partial \omega_n$, $f_{ln} = f_l - f_n$, $\omega_{ln} = \omega_l - \omega_n$, and $Z_{ln}(\tilde{\omega}) = 1/(\omega_{ln}^2 - \tilde{\omega}^2)$. For the intraband identities, we used

$$\frac{f_{nn}(\mathbf{q})}{\omega_{nn}(\mathbf{q})} = f_n' + \mathcal{O}(q^2), \tag{15}$$

which follows from $\omega_{nn}(\mathbf{q}) = (\partial_a \omega_n) q_a + \mathcal{O}(q^3)$ and $f_{nn}(\mathbf{q}) = (\partial_a f_n) q_a + \mathcal{O}(q^3)$, where $\partial_a = \partial_{k_a}$.

### 3.3 Expansion in q of the optical matrix elements

#### 3.3.1 Nondegenerate bands

Energy eigenstates are only defined up to overall phase factors, and observable quantities cannot depend on this phase arbitrariness. In the case of nondegenerate Bloch bands, physical observables must remain invariant under single-band quantum-mechanical "gauge transformations" of the form $|u_n\rangle \rightarrow e^{-i\beta_n}|u_n\rangle$, where $\beta_n$ is a real function of $\mathbf{k}$.

As the $\mathcal{M}$ matrix defined by Eqs. (8) and (9) is clearly gauge invariant, the same must be true for its expansion coefficients entering Eq. (13) for $\sigma_{ab,c}(\tilde{\omega})$. When evaluating those coefficients, we would like to insist that each individual contribution – and not just their sum – is gauge invariant. Doing so will lead to a physically transparent and numerically robust expression for $\sigma_{ab,c}(\tilde{\omega})$ in terms of origin-independent quantities.

The coefficient $\mathcal{M}^{ab}(\mathbf{0})$ appearing in the first term of Eq. (13) is trivially gauge invariant, as it involves a single term,

$$\mathcal{M}_{nl}^{ab}(\mathbf{0}) = v_{nl}^a v_{ln}^b, \tag{16}$$

which is a product of gauge-covariant velocity matrix elements (clearly, those matrix elements are also origin independent). Instead, the coefficient $\mathcal{M}^{ab,c}$ appearing in the second term comprises several terms, not all of which are individually gauge invariant. The problematic terms are those that contain matrix elements such as $\langle u_n|v_a|\partial_c u_l\rangle$, because Bloch-state derivatives transform noncovariantly as $|\partial_\mathbf{k} u_n\rangle \rightarrow e^{-i\beta_n}(|\partial_\mathbf{k} u_n\rangle - i(\partial_\mathbf{k}\beta_n)|u_n\rangle)$. This can be fixed by writing $|\partial_\mathbf{k} u_n\rangle$ as $|\mathbf{D}_\mathbf{k} u_n\rangle - i\mathcal{A}_n|u_n\rangle$, where $|\mathbf{D}_\mathbf{k} u_n\rangle$ is the covariant derivative [39] and $\mathcal{A}_n = \langle u_n|i\partial_\mathbf{k} u_n\rangle$ is the intraband Berry connection. The terms containing Berry connections cancel out, leaving

$$\mathcal{M}_{nl}^{ab,c} = \frac{1}{2}\Big(v_{nl}^a\langle D_c u_l|v_b|u_n\rangle + \langle u_n|v_a|D_c u_l\rangle v_{ln}^b - \langle D_c u_n|v_a|u_l\rangle v_{ln}^b - v_{nl}^a\langle u_l|v_b|D_c u_n\rangle\Big)$$
$$+ \frac{ig_s}{2m_e}\Big(\epsilon_{acd}S_{nl}^d v_{ln}^b - \epsilon_{bcd}S_{ln}^d v_{nl}^a\Big), \tag{17}$$

where every term is a gauge-invariant product of gauge-covariant matrix elements, just like in Eq. (16).

#### 3.3.2 Degenerate bands

Gyrotropic birefringence and nonreciprocal directional dichroism occur in antiferromagnetic crystals such as $Cr_2O_3$ [3, 4], where the energy bands are doubly degenerate at every $\mathbf{k}$ as a result of the combined $\mathcal{PT}$ symmetry [48]. To treat such cases, we introduce degeneracy indices $\lambda$ and $\nu$ for the Bloch states in bands $l$ and $n$, respectively. The Kubo formula (11) remains unchanged, but the matrix element therein becomes a trace over the degeneracy indices, $\mathcal{M}_{nl}^{ab} = \sum_{\lambda,\nu}(I_{l\lambda,n\nu}^a)^*I_{l\lambda,n\nu}^b$. The reasoning leading up to Eq. (17) follows through, provided that the covariant derivative is generalized as $|\mathbf{D}_\mathbf{k} u_{n\nu}\rangle = |\partial_\mathbf{k} u_{n\nu}\rangle + i\sum_{\nu'}|u_{n\nu'}\rangle\mathcal{A}_n^{\nu'\nu}$, where $\mathcal{A}_n^{\nu'\nu} = \langle u_{n\nu'}|i\partial_\mathbf{k} u_{n\nu}\rangle$ [39]. The object $|\mathbf{D}_\mathbf{k} u_{n\nu}\rangle$ transforms covariantly under multiband gauge transformations of the form $|u_{n\nu}\rangle \rightarrow \sum_{\nu'}|u_{n\nu'}\rangle U_n^{\nu'\nu}$, where $U_n$ is a $\mathbf{k}$-dependent unitary matrix in the degeneracy indices. To alleviate the notation, from now on we will assume nondegenerate bands.

### 3.4 Conversion to length (multipole) form

As we started out from the Kubo formula in the velocity gauge, the optical matrix elements (16) and (17) entering Eq. (13) for $\sigma_{ab,c}(\tilde{\omega})$ are written in terms of the velocity operator. Now, we would like to recast those matrix elements in a "length form" that brings out their

multiple character. In the case of molecules [9, 49], this is achieved by means of identities such as $\langle\phi_l|\mathbf{v}|\phi_n\rangle = i\omega_{ln}\langle\phi_l|\mathbf{r}|\phi_n\rangle$.

In periodic crystals, where the velocity operator is given by the gradient of the Hamiltonian, the conversion from velocity to length form follows from the identity

$$(\partial_{\mathbf{k}}H)|u_n\rangle = (\partial_{\mathbf{k}}\varepsilon_n)|u_n\rangle - (H - \varepsilon_n)|\mathbf{D}_{\mathbf{k}}u_n\rangle, \tag{18}$$

which can be obtained by differentiating $H|u_n\rangle = \varepsilon_n|u_n\rangle$, and then writing $|\partial_{\mathbf{k}}u_n\rangle$ as $|\mathbf{D}_{\mathbf{k}}u_n\rangle - i\mathcal{A}_n|u_n\rangle$. Contracting with $\langle u_l|$ gives $\mathbf{v}_{ln} = \delta_{ln}\mathbf{v}_n + i\omega_{ln}\mathbf{A}_{ln}$, where $\mathbf{v}_n$ is the band velocity, and

$$\mathbf{A}_{ln} = \langle u_l|i\mathbf{D}u_n\rangle = (1 - \delta_{ln})\langle u_l|i\partial_{\mathbf{k}}u_n\rangle = \begin{cases} \frac{\mathbf{v}_{ln}}{i\omega_{ln}}, & \text{if } l \neq n, \\ 0, & \text{if } l = n, \end{cases} \tag{19}$$

is the interband Berry connection.

With Eqs. (18) and (19), one can split Eq. (17) for $\mathcal{M}_{nl}^{ab,c}$ as

$$\mathcal{M}_{nl}^{ab,c} = i\left(v_{nl}^a B_{ln}^{bc} - v_{ln}^b B_{nl}^{ac}\right) + \frac{\omega_{ln}}{2}\left[(v_n^a + v_l^a)A_{ln}^b A_{nl}^c + (v_n^b + v_l^b)A_{nl}^a A_{ln}^c\right], \tag{20}$$

where we have defined

$$B_{ln}^{bc} = \frac{1}{2i\hbar}\left(\langle D_b u_l|H - \varepsilon_l|D_c u_n\rangle - \langle D_c u_l|H - \varepsilon_n|D_b u_n\rangle\right) - \frac{g_s}{2m_e}\epsilon_{bcd}S_{ln}^d. \tag{21}$$

The first term in Eq. (20) is molecular for $l \neq n$ and band dispersive for $l = n$, whereas the second term is purely band dispersive and vanishes for $l = n$ (the distinction between molecular and band-dispersive contributions was introduced in Sec. 3.1).

From the gauge-covariant and Hermitian matrices $A^a$ and $B^{bc}$, we can now define for crystals the intrinsic multipole transition moments that were introduced in Eq. (2) for molecules. The intrinsic electric-dipole matrix is $\bar{d}^a = -eA^a$, while the intrinsic magnetic-dipole and electric-quadrupole matrices are related to the antisymmetric and symmetric parts of $B^{bc}$ as follows,

$$\bar{m}_{ln}^a = \frac{e}{2}\epsilon_{abc}B_{ln}^{bc}, \tag{22a}$$

$$\bar{q}_{ln}^{bc} = \frac{ie}{\omega_{ln}}\left(B_{ln}^{bc} + B_{ln}^{cb}\right). \tag{22b}$$

Thus,

$$\bar{\mathbf{d}}_{nl} = -e\langle u_n|i\mathbf{D}_{\mathbf{k}}u_l\rangle, \tag{23a}$$

$$\bar{\mathbf{m}}_{ln} = \frac{e}{2i\hbar}\langle\mathbf{D}_{\mathbf{k}}u_l|\times\left(H - \frac{\varepsilon_l + \varepsilon_n}{2}\right)|\mathbf{D}_{\mathbf{k}}u_n\rangle - \frac{eg_s}{2m_e}\mathbf{S}_{ln}, \tag{23b}$$

$$\bar{q}_{ln}^{bc} = -\frac{e}{2}\left(\langle D_b u_l|D_c u_n\rangle + \langle D_c u_l|D_b u_n\rangle\right), \tag{23c}$$

where $\bar{\mathbf{m}}_{ln}$ comprises orbital and spin contributions. By expanding the covariant derivatives and then setting $l = n$, one finds that $\bar{\mathbf{d}}_{nn} = \mathbf{0}$ [see Eq. (19)], and that $\bar{\mathbf{m}}_{nn}$ and $-\bar{q}_{nn}^{bc}/e$ are respectively the intrinsic magnetic moment $\mathbf{m}_n$ [40] and the quantum metric $g_n^{bc}$ [41] of a Bloch eigenstate.

Using Eq. (19) for $\mathbf{A}_{ln}$ together with the completeness relation, Eq. (23) can be recast in a more convenient form for numerical work,

$$\bar{\mathbf{d}}_{nl} = \begin{cases} ie\dfrac{\mathbf{v}_{nl}}{\omega_{nl}}, & \text{if } l \neq n, \\ 0, & \text{if } l = n, \end{cases} \tag{24a}$$

$$\bar{\mathbf{m}}_{ln}^{\text{orb}} = \frac{e}{4i} \sum_{p \neq l,n} \left( \frac{1}{\omega_{pl}} + \frac{1}{\omega_{pn}} \right) \mathbf{v}_{lp} \times \mathbf{v}_{pn}, \tag{24b}$$

$$\bar{q}_{ln}^{bc} = -\frac{e}{2} \sum_{p \neq l,n} \left[ \frac{v_{lp}^b v_{pn}^c}{\omega_{pl} \omega_{pn}} + (b \leftrightarrow c) \right]. \tag{24c}$$

As they are written in terms of matrix elements of the velocity operator, these expressions are manifestly origin independent. The correspondence with the molecular expressions in Eq. (2) will be established in Sec. 4.

In the case of degenerate bands, Eq. (23) gets modified in the manner described in Sec. 3.3.2. The modified Eq. (24) remains nonsingular, as its energy denominator only contains energy differences between nondegenerate bands.

There is at present considerable interest in quantum-geometric quantities associated with interband optical responses [50]. In this regard, we note that the quantity $-\bar{q}_{ln}^{bc}/e$ is distinct from the "band-resolved quantum metric" $g_{ln}^{bc} = \left( A_{ln}^b A_{nl}^c + A_{ln}^c A_{nl}^b \right)/2$ that has been introduced in connection with nonlinear optical responses [51–53]. The quantity $g_{ln}^{bc}$ is gauge invariant for every $l$ and $n$, and when summed over $l$ it gives the quantum metric of band $n$, $g_n^{bc} = \text{Re} \langle \partial_b u_n | \partial_c u_n \rangle - \mathcal{A}_n^b \mathcal{A}_n^c$. Instead, $-\bar{q}_{ln}^{bc}/e$ is gauge covariant for $l \neq n$, and for $l = n$ it reduces to $g_n^{bc}$.

## 3.5 Final expression

We have now gathered all the needed ingredients to evaluate Eq. (13) for $\sigma_{ab,c}(\tilde{\omega})$, namely the expansion coefficients of $\mathcal{E}$ in Eq. (14), and those of $\mathcal{M}$ in Eqs. (16) and (20). In Eqs. (26) and (31) below, we break down the resulting expression into antisymmetric ($\mathcal{T}$-even) and symmetric ($\mathcal{T}$-odd) parts. The real and imaginary parts of those two equations are either absorptive or reactive, as per Eq. (5).

To arrive at Eqs. (26) and (31), several terms containing double band summations were eliminated by exchanging the $l$ and $n$ indices (note also that the $l = n$ terms therein vanish, because $f_{nn} = \omega_{nn} = A_{nn}^a = 0$). Those equations are written in terms of the $A^a$ and $B^{bc}$ matrices, which in turn are related to the intrinsic multipole transition moments by

$$A_{nl}^a = -\frac{1}{e}\bar{d}_{nl}^a, \quad B_{ln}^{bc} = \frac{1}{e}\bar{m}_{ln}^a \epsilon_{abc} + \frac{\omega_{ln}}{2ie}\bar{q}_{ln}^{bc}. \tag{25}$$

According to Eq. (24), the $A^a$ and $B^{ab}$ matrices depend exclusively on band energies and interband velocity matrix elements. From the restrictions imposed on these quantities by the presence of $\mathcal{P}$, $\mathcal{T}$, or $\mathcal{P}\mathcal{T}$ symmetry [48], the restrictions on $A^a$ and $B^{ab}$ can be deduced. In this way, it may be verified that the expressions given below satisfy the symmetry constraints discussed at the end of Sec. 2.

### 3.5.1 Antisymmetric (time-even) part

The antisymmetric part of $\sigma_{ab,c}(\tilde{\omega})$ takes the form

$$
\begin{aligned}
\frac{\hbar}{e^2}\sigma_{ab,c}^{\mathrm{A}}(\tilde{\omega}) = \tilde{\omega}\sum_{n,l}\int_{\mathbf{k}} Z_{ln}(\tilde{\omega})\Big\{ &-f_{ln}\mathrm{Im}\big[A_{nl}^a B_{ln}^{bc} - (a \leftrightarrow b)\big] \\
&+ f_{ln}\Big[\frac{1}{2}\big(v_n^a + v_l^a\big)\mathrm{Im}\big(A_{nl}^b A_{ln}^c\big) - (a \leftrightarrow b)\Big] \\
&+ f_{ln}\big(3\omega_{ln}^2 - \tilde{\omega}^2\big)Z_{ln}(\tilde{\omega})\mathrm{Im}\big(A_{nl}^a A_{ln}^b\big)\frac{1}{2}\big(v_n^c + v_l^c\big) \\
&- f_n'\omega_{ln}\mathrm{Im}\big(A_{nl}^a A_{ln}^b\big)v_n^c\Big\} \\
&+ \frac{1}{\tilde{\omega}}\sum_n\int_{\mathbf{k}} f_n'\big(v_n^a B_{nn}^{bc} - v_n^b B_{nn}^{ac}\big) .
\end{aligned}
\tag{26}
$$

The five terms in this expression can be classified as follows. The first is molecular, while the others are band dispersive; the four inside curly brackets are interband, while the fifth is intraband; and the first three are Fermi-sea-like, while the last two are Fermi-surface-like.

In insulators and cold semiconductors, only the Fermi-sea terms survive, and one can compare with previous treatments of optical activity in nonconducting crystals. In Refs. [19, 22], the sole band-dispersion contribution to $\sigma_{ab,c}^{\mathrm{A}}(\tilde{\omega})$ came from differentiating the $\mathcal{E}$ matrix, that is, from the first term of Eq. (13) for $\sigma_{ab,c}(\tilde{\omega})$. It went unnoticed in those works that the other term in that equation – where one differentiates the $\mathcal{M}$ matrix instead – is not purely molecular, as shown in Eq. (20). This is why we have not two but three Fermi-sea terms in Eq. (26), one molecular and two band dispersive.

In conductors, the Fermi-surface terms contribute as well. Using Eq. (25), the last term in Eq. (26) becomes $(\epsilon_{acd}K_{bd} - \epsilon_{bcd}K_{ad})/(e\tilde{\omega})$, where

$$
K_{ab} = -\sum_n\int_{\mathbf{k}} f_n' v_n^a m_n^b = \sum_n\int_{\mathbf{k}} f_n \partial_a m_n^b .
\tag{27}
$$

This intraband contribution to optical activity involving the intrinsic magnetic moment of conduction electrons was identified in Refs. [23, 24], and was evaluated for $p$-doped tellurium in Refs. [28, 54]. The fourth term in Eq. (26) gives an additional interband contribution to the optical activity of conductors that was overlooked in previous works.

The low-frequency behavior of the optical rotatory dispersion is different in insulators and in conductors. For simplicity, let us consider the propagation of light along the optical axis $z$ of a uniaxial crystal. The rotatory power is given by [1]

$$
\rho(\omega, \tau) = \frac{\omega}{2c^2\epsilon_0}\mathrm{Re}\big[\sigma_{xy,z}^{\mathrm{A}}(\omega + i\tau^{-1})\big] ,
\tag{28}
$$

where $\epsilon_0$ is the vacuum permittivity and $c$ is the speed of light. To deal with absorption, the positive infinitesimal $\eta$ in $\tilde{\omega} = \omega + i\eta$ has been reinterpreted heuristically as a phenomenological scattering rate $\tau^{-1}$ [6, 46, 55]. For frequencies and scattering rates well below the threshold for interband transitions, $\omega, \tau^{-1} \ll \omega_{\mathrm{gap}}$, Eq. (26) yields

$$
\rho(\omega, \tau) = \frac{(\omega\tau)^2}{1 + (\omega\tau)^2}a + b\omega^2 .
\tag{29}
$$

The coefficient $b$ comes from the interband terms which have $\tilde{\omega}$ prefactors, and $a = -(e/c^2\epsilon_0\hbar)K_{xx}$ (with $K_{xx} = K_{yy}$) comes from the intraband term with a $1/\tilde{\omega}$ prefactor.

In insulators the coefficient $a$ vanishes, and hence the rotatory power displays the familiar $\omega^2$ dependence at low frequencies [6]; in conductors that coefficient is nonzero, and one can distinguish two different regimes as follows,

$$\rho(\omega, \tau) \simeq \begin{cases} (\tau^2 a + b)\omega^2, & \text{if } \omega\tau \ll 1, \\ a + b\omega^2, & \text{if } \omega\tau \gg 1. \end{cases} \tag{30}$$

In Sec. 6, we will illustrate these low-frequency profiles for a concrete tight-binding model.

### 3.5.2 Symmetric (time-odd) part

The symmetric part of $\sigma_{ab,c}(\tilde{\omega})$ reads

$$
\begin{aligned}
\frac{\hbar}{e^2}\sigma^{\text{S}}_{ab,c}(\tilde{\omega}) = & i\sum_{n,l}\int_{\mathbf{k}} Z_{ln}(\tilde{\omega})\Bigg\{ f_{ln}\omega_{ln}\text{Re}\left[A^a_{nl}B^{bc}_{ln} + (a \leftrightarrow b)\right] \\
& + f_{ln}\omega_{ln}\left[\frac{1}{2}\left(v^a_n + v^a_l\right)\text{Re}\left(A^b_{nl}A^c_{ln}\right) + (a \leftrightarrow b)\right] \\
& - f_{ln}\omega^3_{ln}Z_{ln}(\tilde{\omega})\text{Re}\left(A^a_{nl}A^b_{ln}\right)\left(v^c_n + v^c_l\right) \\
& + f'_n\omega^2_{ln}\text{Re}\left(A^a_{nl}A^b_{ln}\right)v^c_n\Bigg\} \\
& - \frac{i}{\tilde{\omega}^2}\sum_n\int_{\mathbf{k}} f'_n v^a_n v^b_n v^c_n.
\end{aligned}
\tag{31}
$$

The first three terms are Fermi-sea-like, and can be compared with the expressions obtained for insulators in Ref. [22]. The third is band dispersive, and it corresponds to Eq. (31) in that work, while the first two add up to Eq. (30) therein, revealing its mixed molecular/dispersive character.

The remaining two terms in Eq. (31) are Fermi-surface-like, and they can be compared with the expressions obtained for metals in Ref. [25]. The last term was identified in that work. Writing $\omega^2_{ln}Z_{ln}(\tilde{\omega})$ as $1 + \tilde{\omega}^2 Z_{ln}(\tilde{\omega})$ and noting that $\text{Re}\sum_l A^a_{nl}A^b_{ln}$ is the quantum metric $g^{ab}_n = -\bar{q}^{ab}_{nn}/e$ [this can be seen from Eqs. (19) and (24c)], the fourth term in Eq. (31) splits into intraband and interband parts as follows,

$$
i\sum_n\int_{\mathbf{k}} f'_n g^{ab}_n v^c_n + i\tilde{\omega}^2\sum_{n,l}\int_{\mathbf{k}} f'_n Z_{ln}(\tilde{\omega})\text{Re}\left(A^a_{nl}A^b_{ln}\right)v^c_n. \tag{32}
$$

The intraband piece is similar to the quantity $K_{ab}$ in Eq. (27), but with the intrinsic magnetic moment replaced by the quantum metric (intrinsic quadrupole moment). An equivalent result was obtained in Ref. [25] for two-band models, but without invoking the identities leading up to Eq. (32), which is what allowed us to isolate a quantum-metric contribution in the general multiband case.

## 3.6 Magnetoelectric and quadrupolar responses

As recognized in Ref. [3], the physical basis for $\sigma_{ab,c}(\omega)$ is provided by quadrupolar and magnetoelectric couplings. The quadrupolar couplings are described by a $\mathcal{T}$-odd totally-symmetric tensor $\gamma_{abc}(\omega)$, and the magnetoelectric ones by a $\mathcal{T}$-odd traceless tensor $\tilde{\alpha}_{ab}(\omega)$ together with

a $\mathcal{T}$-even tensor $\breve{\alpha}_{ab}(\omega)$. Those tensors are defined as [3, 22]

$$\gamma_{abc}(\omega) = \frac{1}{3i}\left[\sigma^{\mathrm{S}}_{ab,c}(\omega) + \sigma^{\mathrm{S}}_{bc,a}(\omega) + \sigma^{\mathrm{S}}_{ca,b}(\omega)\right], \tag{33a}$$

$$\tilde{\alpha}_{ab}(\omega) = \frac{1}{3i}\sigma^{\mathrm{S}}_{ac,d}(\omega)\epsilon_{cdb}, \tag{33b}$$

$$\breve{\alpha}_{ab}(\omega) = \frac{1}{4i}\epsilon_{bcd}\left[\sigma^{\mathrm{A}}_{cd,a}(\omega) - 2\sigma^{\mathrm{A}}_{ac,d}(\omega)\right], \tag{33c}$$

so that

$$\sigma^{\mathrm{S}}_{ab,c}(\omega) = i\left[\epsilon_{acd}\tilde{\alpha}_{bd}(\omega) + \epsilon_{bcd}\tilde{\alpha}_{ad}(\omega)\right] + i\gamma_{abc}(\omega), \tag{34a}$$

$$\sigma^{\mathrm{A}}_{ab,c}(\omega) = i\left[\epsilon_{acd}\breve{\alpha}_{bd}(\omega) - \epsilon_{bcd}\breve{\alpha}_{ad}(\omega)\right]. \tag{34b}$$

Inserting Eq. (34) in Eq. (3) for the induced current density and then using the Maxwell-Faraday equation yields the constitutive relation

$$j^{\omega,\mathbf{q}}_a = (\boldsymbol{\nabla}\times\mathbf{M}^{\omega,\mathbf{q}})_a + \left[\tilde{\alpha}_{ab}(\omega) - \breve{\alpha}_{ab}(\omega)\right]\partial_t B^{\omega,\mathbf{q}}_b + \gamma_{abc}(\omega)\nabla_b E^{\omega,\mathbf{q}}_c, \tag{35}$$

where $\mathbf{X}^{\omega,\mathbf{q}}$ denotes $\mathbf{X}(\omega,\mathbf{q})e^{i(\mathbf{q}\cdot\mathbf{r}-i\omega t)}$, and $M^{\omega,\mathbf{q}}_a = \left[\tilde{\alpha}_{ba}(\omega) + \breve{\alpha}_{ba}(\omega)\right]E^{\omega,\mathbf{q}}_b$.

In the quasi-static limit the electric and magnetic fields become decoupled, and the three terms in Eq. (35) describe separate physical responses. The first corresponds to a magnetization current induced by a uniform electric field; the second to a current induced by a time-varying magnetic field; and the third to a current induced by a spatially-varying electric field. The first two are direct and inverse magnetoelectric effects [1], and the third is an electric quadrupolar effect.

The $\mathcal{T}$-even magnetoelectric tensor $\breve{\alpha}_{ab}$ describes a "kinetic" magnetoelectric effect in gyrotropic conductors [56], while the $\mathcal{T}$-odd tensor $\tilde{\alpha}_{ab}$ describes magnetoelectric effects in both insulators and conductors. In the case of insulators, the inverse magnetoelectric response can be expressed as as a polarization current, $\partial_t P^{\omega,\mathbf{q}}_a = \tilde{\alpha}_{ab}(\omega)\partial_t B^{\omega,\mathbf{q}}_b$; integrating the adiabatic current in the quasi-static limit, one obtains the familiar form $P_a = \tilde{\alpha}_{ab}B_b$ of the inverse magnetoelectric effect [1]. The full $\mathcal{T}$-odd magnetoelectric tensor includes an additional trace ("axion") piece $(\theta e^2/2\pi h)\delta_{ab}$; the axion angle $\theta$ is boundary sensitive [39], and this is why it is not captured by the present formalism, which is based on the electromagnetic response of a bulk medium.

Quantum-mechanical expressions for the static magnetoelectric and quadrupolar susceptibilities can be obtained by inserting in Eq. (33) the formulas given in Sec. 3.5 for $\sigma^{\mathrm{S}}_{ab,c}(\omega)$ and $\sigma^{\mathrm{A}}_{ab,c}(\omega)$, evaluated at $\omega = 0$. This has been done in previous works for selected terms only: in Ref. [22], the Fermi-sea orbital contribution to $\tilde{\alpha}_{ab}(0)$ was shown to give the traceless part of the orbital magnetoelectric susceptibility tensor of insulators [57]; and in Ref. [23], the intraband contribution to $\breve{\alpha}_{ab}(0)$ was shown to reproduce the kinetic magnetoelectric susceptibility obtained from Boltzmann transport in the relaxation time approximation combined with the modern theory of orbital magnetization [58].

Compared to those previous works, the present formalism provides a more complete description. In particular, it captures $\mathcal{T}$-odd magnetoelectric and electric-quadrupolar effects in conductors that so far have only been treated semiclassically [25, 59–61], and which were found to involve the quantum metric. A detailed account of magnetoelectric and electric-quadrupolar responses on the basis of the present formalism will be given in a separate work.

## 4  Molecular limit

In this section, we analyze the molecular limit of our formalism. First, we show how in that limit the bulk expressions for the intrinsic transition moments $\bar{d}$, $\bar{m}$, and $\bar{q}$ reduce to those in

Eq. (2). We then show how the formulas for $\sigma^{\text{A}}_{ab,c}(\tilde{\omega})$ and $\sigma^{\text{S}}_{ab,c}(\tilde{\omega})$ reduce to the standard molecular expressions in terms of the ordinary transition moments $d$, $m$, and $q$ in Eq. (1).

Consider an idealized molecular crystal composed of nonoverlapping units. For such a crystal, the cell-periodic Bloch states assume the form [62, 63]

$$u_{n\mathbf{k}}(\mathbf{r}) \doteq e^{-i\mathbf{k}\cdot\xi(\mathbf{r})}\phi_n[\xi(\mathbf{r})], \tag{36}$$

where $\xi(\mathbf{r}) = \mathbf{r} - \mathbf{R}(\mathbf{r})$ is the intracell coordinate, $\mathbf{R}(\mathbf{r})$ is the lattice vector that folds the absolute coordinate $\mathbf{r}$ into the home unit cell, $\phi_n(\mathbf{r})$ is vanishingly small outside that cell, and $\doteq$ denotes an equality that only holds in the molecular limit. The intraband Berry connection can now be easily evaluated by integrating over the home cell,

$$\mathcal{A}_n \doteq \int_{\text{cell}} d\mathbf{r}\, \phi_n^*(\mathbf{r})e^{i\mathbf{k}\cdot\mathbf{r}}\, i\partial_{\mathbf{k}}\left[e^{-i\mathbf{k}\cdot\mathbf{r}}\phi_n(\mathbf{r})\right] = \bar{\mathbf{r}}_n\,, \tag{37}$$

and the covariant derivative of a Bloch state reduces to

$$\mathbf{D}_{\mathbf{k}}u_{n\mathbf{k}}(\mathbf{r}) \doteq -ie^{-i\mathbf{k}\cdot\mathbf{r}}(\mathbf{r} - \bar{\mathbf{r}}_n)\,\phi_n(\mathbf{r})\,, \tag{38}$$

for $\mathbf{r}$ in the home cell. Using this identity in Eq. (23) for $\bar{d}$, $\bar{m}$, and $\bar{q}$, we recover after some manipulations the expressions in Eq. (2). In the case of $\bar{m}$, it is necessary to invoke the operator identity $[v_a, r_b] = [v_b, r_a]$ to rewrite $(\mathbf{r} \times \mathbf{v} - \mathbf{v} \times \mathbf{r})/2$ as $\mathbf{r} \times \mathbf{v}$.

With these relations in hand, we can address the molecular limit of Eqs. (26) and (31) for $\sigma^{\text{A}}_{ab,c}(\tilde{\omega})$ and $\sigma^{\text{S}}_{ab,c}(\tilde{\omega})$. Since the energy bands become dispersionless in that limit, all band-dispersion terms in those equations vanish, leaving only the first term in each of them; and since the transition moments also become independent of $\mathbf{k}$, we can set $\int_{\mathbf{k}} \doteq 1/V_c$ in those terms ($V_c$ is the cell volume) to find

$$V_c\sigma^{\text{A}}_{ab,c} \doteq \bar{G}'_{ad}\epsilon_{dbc} + \frac{\tilde{\omega}}{2}\bar{a}_{abc} - (a \leftrightarrow b)\,, \tag{39a}$$

$$iV_c\sigma^{\text{S}}_{ab,c} \doteq -\bar{G}_{ad}\epsilon_{dbc} + \frac{\tilde{\omega}}{2}\bar{a}'_{abc} + (a \leftrightarrow b)\,, \tag{39b}$$

where we dropped the $\tilde{\omega}$ dependence for brevity, and defined the (extensive) molecular tensors

$$\bar{G}_{ab} = \frac{1}{\hbar}\sum_{n,l} f_{nl}\omega_{ln}Z_{ln}\text{Re}\left(\bar{d}^a_{nl}\bar{m}^b_{ln}\right)\,, \tag{40a}$$

$$\bar{G}'_{ab} = -\frac{1}{\hbar}\sum_{n,l} f_{nl}\tilde{\omega}Z_{ln}\text{Im}\left(\bar{d}^a_{nl}\bar{m}^b_{ln}\right)\,, \tag{40b}$$

$$\bar{a}_{abc} = \frac{1}{\hbar}\sum_{n,l} f_{nl}\omega_{ln}Z_{ln}\text{Re}\left(\bar{d}^a_{nl}\bar{q}^{bc}_{ln}\right)\,, \tag{40c}$$

$$\bar{a}'_{abc} = -\frac{1}{\hbar}\sum_{n,l} f_{nl}\frac{\omega^2_{ln}}{\tilde{\omega}}Z_{ln}\text{Im}\left(\bar{d}^a_{nl}\bar{q}^{bc}_{ln}\right)\,. \tag{40d}$$

Writing $f_{nl}$ as $f_n(1-f_l) - f_l(1-f_n)$, the $f_{nl}$ factors in the expressions above can be replaced with $2f_n(1-f_l)$. In that form, $\bar{G}$, $\bar{G}'$ and $\bar{a}$ become single-particle versions of the multipolar susceptibility tensors $G$, $G'$ and $a$ defined in Eqs. (2.83), (2.85) and (2.86) of Ref. [7], with one difference: the ordinary transition moments $d$, $m$, and $q$ have been replaced with $\bar{d}$, $\bar{m}$, and $\bar{q}$. It is not immediately clear that the same is true for $\bar{a}'$, since Eq. (40d) contains a factor of $\omega^2_{ln}/\tilde{\omega}$ in place of the $\tilde{\omega}$ factor appearing in Eq. (2.84) of Ref. [7] for $a'$. However, those factors are interchangeable in the expression for $a'$, as can be seen in the manner described around

around Eqs. (2.75–2.78) of Ref. [7]. Thus, $(\bar{G}, \bar{G}', \bar{a}, \bar{a}')$ are origin-independent versions of the molecular tensors $(G, G', a, a')$ entering the standard multipole theory.

Next, let us consider the propagation of light inside our idealized molecular crystal. For a given propagation direction $\hat{\mathbf{n}}$, we define (intensive) optical-activity and gyrotropic-birefringence tensors as $\beta^{A}_{ab}(\hat{\mathbf{n}}) = -\sigma^{A}_{ab,c}\hat{n}_c$ and $\beta^{S}_{ab}(\hat{\mathbf{n}}) = i\sigma^{S}_{ab,c}\hat{n}_c$, respectively. Using Eq. (39), we obtain Eqs. (5.8) and (5.9) of Ref. [7] for those tensors, but with $(\bar{G}, \bar{G}', \bar{a}, \bar{a}')$ in place of $(G, G', a, a')$. Inserting Eq. (2) in Eq. (40), the terms containing the orbital centers drop out from the combinations of molecular tensors appearing in Eq. (39). Thus, $(\bar{d}, \bar{m}, \bar{q})$ can be safely replaced by $(d, m, q)$ for the purpose of evaluating the optical properties of an idealized molecular crystal. This completes the proof that our formalism correctly reduces to the standard single-particle multipole theory in the molecular limit.

In summary, we have in Eqs. (39) and (40) a reformulation of the molecular multipole theory of optical spatial dispersion at linear order in $\mathbf{q}$ in terms of translationally-invariant property tensors. This is in contrast to the standard formulation, where translational invariance is achieved by a cancellation between the origin dependences of the magnetic-dipole and electric-quadrupole terms [6–9].

## 5 Sum rules

In Sec. 3.1, we wrote two alternative Kubo formulas for $\sigma_{ab}(\omega, \mathbf{q})$, namely Eqs. (7) and (11). The former displays apparent $1/\omega$ divergences at $\omega = 0$, whereas the latter is explicitly divergence free. In this section, we scrutinize the mathematical identities that underlie the equivalence between those two formulas at zeroth and first order in $\mathbf{q}$, and relate those identities to the oscillator- and rotatory-strength sum rules.

### 5.1 Equivalence between the two forms of the Kubo formula

Let us denote as $(ie^2/\omega)\Delta_{ab}(\mathbf{q})$ the difference between the reactive parts of the Kubo formulas (7) and (11). Writing $1/[x(a-x)]$ as $(1/a)[1/x + 1/(a-x)]$, we find

$$\Delta_{ab}(\mathbf{q}) = \delta_{ab}\frac{N}{m_e} + \frac{1}{\hbar}\sum_{n,l}\int_{\mathbf{k}}\frac{f_{ln}(\mathbf{q})}{\omega_{ln}(\mathbf{q})}M_{nl}^{ab}(\mathbf{q}), \tag{41}$$

and using

$$\omega_{ln}(\mathbf{q}) = -\omega_{nl}(-\mathbf{q}), \tag{42a}$$

$$f_{ln}(\mathbf{q}) = -f_{nl}(-\mathbf{q}), \tag{42b}$$

$$M_{nl}^{ab}(\mathbf{q}) = \left[M_{ln}^{ab}(-\mathbf{q})\right]^*, \tag{42c}$$

$$M_{nl}^{ab}(\mathbf{q}) = \left[M_{nl}^{ba}(\mathbf{q})\right]^*, \tag{42d}$$

we obtain

$$\operatorname{Re}\Delta_{ab}(\mathbf{q}) = \operatorname{Re}\Delta_{ba}(\mathbf{q}) = \operatorname{Re}\Delta_{ab}(-\mathbf{q}), \tag{43a}$$

$$\operatorname{Im}\Delta_{ab}(\mathbf{q}) = -\operatorname{Im}\Delta_{ba}(\mathbf{q}) = -\operatorname{Im}\Delta_{ab}(-\mathbf{q}). \tag{43b}$$

For the two Kubo formulas to be equivalent, $\Delta_{ab}(\mathbf{q})$ must vanish identically, and according to the derivation in Sec. 3.1 this is guaranteed by the Kramers-Krönig relations. To analyze the behavior of $\Delta_{ab}(\mathbf{q})$ at zeroth and first order in $\mathbf{q}$, we expand it as

$$\Delta_{ab}(\mathbf{q}) = \Delta_{ab}(\mathbf{0}) + \Delta_{ab,c}q_c + \mathcal{O}(q^2). \tag{44}$$

Let us start with the zeroth-order term in the expansion. Writing the electron density $N$ in the first term of Eq. (41) as $\sum_n \int_{\mathbf{k}} f_n$, and using the identity in Eq. (15), followed by an integration by parts, to deal with the $l = n$ contribution to the second term, we obtain

$$\Delta_{ab}(\mathbf{0}) = \sum_n \int_{\mathbf{k}} f_n \left[ \frac{\delta_{ab}}{m_e} + 2 \sum_{l \neq n} \frac{\text{Re}\left(v_{nl}^a v_{ln}^b\right)}{\varepsilon_n - \varepsilon_l} - \frac{1}{\hbar^2} \frac{\partial^2 \varepsilon_n}{\partial k_a \partial k_b} \right] = 0. \tag{45}$$

The quantity in square brackets is formally real and symmetric in accordance with Eq. (43a), and it vanishes identically by virtue of the effective-mass theorem. We note that the same theorem was invoked in Ref. [64] to remove the apparent divergence at $\omega = 0$ in the dielectric function of insulators and semiconductors.

In preparation for analyzing the first-order term in Eq. (44), let us compare Eq. (41) for $\Delta_{ab}(\mathbf{q})$ with the $\omega \to 0$ limit of $\Pi_{ab}(\omega, \mathbf{q}) = i\omega \sigma_{ab}(\omega, \mathbf{q})$ [see Eq. (3)], evaluated using the Kubo formula (7). This gives $-e^2 \Delta_{ab}(\mathbf{q}) = \Pi_{ab}(0, \mathbf{q})$, and so $-e^2 \Delta_{ab,c} = \partial_{q_c} \Pi_{ab}(0, \mathbf{q})\big|_{\mathbf{q}=0}$. From the analysis of this quantity in Ref. [23] (see Sec. III.A.1 of its Supplemental Material), we conclude that

$$\Delta_{ab,c} = -\frac{i}{\hbar} \epsilon_{abc} \sum_n \int_{\mathbf{k}} f_n \mathbf{v}_n \cdot \mathbf{\Omega}_n = 0, \tag{46}$$

where $\mathbf{\Omega}_n$ is the Berry curvature. In agreement with Eq. (43b), the expression above is purely imaginary and antisymmetric in $a$ and $b$. It vanishes identically for topological reasons, and that amounts to a no-go theorem for the chiral magnetic effect in equilibrium [23].

In a recent work [36], an expression was derived for $\sigma_{ab,c}(\omega)$ that contains a term diverging as $1/\omega$. The authors found that the prefactor of that term was neglibible for a specific tight-binding model, but they were unable to confirm analytically that it vanishes in general. The removal of that apparent divergence can be achieved by means of Eq. (46).

The vanishing of $\Delta_{ab}(\mathbf{q})$ dictates the high-frequency behavior of the optical conductivity as follows. Suppose there is a frequency $\omega_{\text{max}}$ above which the system does not absorb [6]; setting $\omega \gg \omega_{\text{max}}$ in Eqs. (11) and (12) and comparing with Eq. (41), we can deduce that

$$\sigma_{ab}(\omega \gg \omega_{\text{max}}, \mathbf{q}) = \delta_{ab} \frac{ie^2 N}{\omega m_e} - \frac{ie^2}{\omega} \Delta_{ab}(\mathbf{q}). \tag{47}$$

Thus, at high frequencies the optical conductivity reduces to the diamagnetic term; and since that term is independent of $\mathbf{q}$, we conclude that $\sigma_{ab}(\omega, \mathbf{0})$ decays as $1/\omega$.

In Sec. 5.2.4 of Ref. [6], the high-frequency behavior of the optical activity of molecules was inferred from the rotatory-strength sum rule. This is consistent with the present analysis, because that sum rule is a direct consequence of the vanishing of $\Delta_{ab,c}$, as we will now show.

## 5.2 Optical sum rules

Consider the sum rules obtained by integrating over positive frequencies the absorptive part of the optical conductivity, taking into account both interband and intraband absorption. Writing $\int_0^\infty f(\omega) d\omega$ as $\langle f(\omega) \rangle$ and using Eq. (5) yields

$$\left\langle \sigma_{ab}^{\text{H}}(\omega, \mathbf{q}) \right\rangle = \left\langle \text{Re}\, \sigma_{ab}^{\text{S}}(\omega, \mathbf{q}) \right\rangle + i \left\langle \text{Im}\, \sigma_{ab}^{\text{A}}(\omega, \mathbf{q}) \right\rangle. \tag{48}$$

To evaluate this quantity, we begin by taking the Hermitian part of Eq. (11) for $\eta \to 0^+$,

$$\sigma_{ab}^{\text{H}}(\omega, \mathbf{q}) = -\frac{\pi e^2}{\hbar} \sum_{n,l} \int_{\mathbf{k}} \frac{f_{ln}(\mathbf{q})}{\omega_{ln}(\mathbf{q})} \mathcal{M}_{nl}^{ab}(\mathbf{q}) \delta\left[\omega - \omega_{ln}(\mathbf{q})\right]. \tag{49}$$

Making the substitution

$$f_{ln}(\mathbf{q}) = f_l(\mathbf{k} + \mathbf{q}/2)[1 - f_n(\mathbf{k} - \mathbf{q}/2)] - f_n(\mathbf{k} - \mathbf{q}/2)[1 - f_l(\mathbf{k} + \mathbf{q}/2)], \tag{50}$$

and noting that at zero temperature only the second term contributes to Eq. (49) when $\omega > 0$ and $\mathbf{q} \approx \mathbf{0}$, we obtain

$$\left\langle \operatorname{Re} \sigma_{ab}^{\mathrm{S}}(\omega, \mathbf{q}) \right\rangle + i \left\langle \operatorname{Im} \sigma_{ab}^{\mathrm{A}}(\omega, \mathbf{q}) \right\rangle = R_{ab}(\mathbf{q}) + i\, I_{ab}(\mathbf{q}), \tag{51}$$

where we have defined

$$R_{ab}(\mathbf{q}) = \frac{\pi e^2}{\hbar} \sum_{n,l} \int_{\mathbf{k}} f_n(\mathbf{k} - \mathbf{q}/2)[1 - f_l(\mathbf{k} + \mathbf{q}/2)] \frac{\operatorname{Re}\left[\mathcal{M}_{nl}^{ab}(\mathbf{q})\right]}{\omega_{ln}(\mathbf{q})}, \tag{52a}$$

$$I_{ab}(\mathbf{q}) = \frac{\pi e^2}{\hbar} \sum_{n,l} \int_{\mathbf{k}} f_n(\mathbf{k} - \mathbf{q}/2)[1 - f_l(\mathbf{k} + \mathbf{q}/2)] \frac{\operatorname{Im}\left[\mathcal{M}_{nl}^{ab}(\mathbf{q})\right]}{\omega_{ln}(\mathbf{q})}. \tag{52b}$$

Let us split $R_{ab}(\mathbf{q})$ and $I_{ab}(\mathbf{q})$ in Eq. (51) into even and odd parts in $\mathbf{q}$. Using the identities in Eq. (42), one finds that the even part of $R_{ab}(\mathbf{q})$ plus the odd part of $i I_{ab}(\mathbf{q})$ is proportional to the second term in Eq. (41) for $\Delta_{ab}(\mathbf{q})$. Therefore,

$$\left\langle \operatorname{Re} \sigma_{ab}^{\mathrm{S}}(\omega, \mathbf{q}) \right\rangle = \frac{\pi e^2}{2} \left[ \delta_{ab} N/m_e - \operatorname{Re} \Delta_{ab}(\mathbf{q}) \right] + \frac{1}{2} \left[ R_{ab}(\mathbf{q}) - R_{ab}(-\mathbf{q}) \right], \tag{53a}$$

$$\left\langle \operatorname{Im} \sigma_{ab}^{\mathrm{A}}(\omega, \mathbf{q}) \right\rangle = -\frac{\pi e^2}{2} \operatorname{Im} \Delta_{ab}(\mathbf{q}) + \frac{1}{2} \left[ I_{ab}(\mathbf{q}) + I_{ab}(-\mathbf{q}) \right], \tag{53b}$$

where we keep track of the vanishing quantity $\Delta_{ab}(\mathbf{q})$.

The expansion of Eq. (53) in powers of $\mathbf{q}$ generates a series of sum rules. Since, according to Eq. (43), the terms $\operatorname{Re} \Delta_{ab}$ and $\operatorname{Im} \Delta_{ab}$ only contribute (formally) at even and odd orders in $\mathbf{q}$, respectively, and since the reverse is true for the second terms in Eqs. (53a) and (53b), we obtain

$$\left\langle \operatorname{Re} \sigma_{ab}^{\mathrm{S}}(\omega, \mathbf{0}) \right\rangle = \frac{\pi e^2}{2} \left[ \delta_{ab} N/m_e - \Delta_{ab}(\mathbf{0}) \right], \tag{54a}$$

$$\left\langle \operatorname{Im} \sigma_{ab}^{\mathrm{A}}(\omega, \mathbf{0}) \right\rangle = I_{ab}(\mathbf{0}), \tag{54b}$$

$$\left\langle \operatorname{Re} \sigma_{ab,c}^{\mathrm{S}}(\omega) \right\rangle = R_{ab,c}, \tag{54c}$$

$$\left\langle \operatorname{Im} \sigma_{ab,c}^{\mathrm{A}}(\omega) \right\rangle = -\frac{\pi e^2}{2} \Delta_{ab,c}, \tag{54d}$$

to linear order in $\mathbf{q}$. Below, we consider each of these identities in turn.

Once we set $\Delta_{ab}(\mathbf{0}) = 0$ in accordance with Eq. (45), Eq. (54a) becomes the oscillator-strength sum rule [1]

$$\left\langle \operatorname{Re} \sigma_{ab}^{\mathrm{S}}(\omega, \mathbf{0}) \right\rangle = \frac{\omega_{\mathrm{p}}^2}{8} \delta_{ab}, \tag{55}$$

where $\omega_{\mathrm{p}} = (4\pi e^2 N/m_e)^{1/2}$ is the plasma frequency. As already mentioned, for tight-binding models the diamagnetic term in Eq. (7) changes form while Eq. (11) remains unchanged, which leads to a modified oscillator-strength sum rule [47].

Equation (54b) is the rotatory-strength sum rule for magnetic circular dichroism. At $\mathbf{q} = \mathbf{0}$, the intraband part of Eq. (52b) vanishes because $\mathcal{M}_{nn}^{ab}(\mathbf{0})$ is real, and from the interband part we recover the bulk expression given in Ref. [65] for that sum rule. If a single band is occupied, the integrated magnetic circular dichroism spectrum is proportional to the intrinsic orbital magnetic moment of the Bloch states in that band [65, 66].

Equation (54c) is a sum rule for nonreciprocal directional dichroism. An explicit expression can be obtained by expanding Eq. (52a) to first order in $\mathbf{q}$.

Finally, by setting $\Delta_{ab,c} = 0$ in Eq. (54d) in accordance with Eq. (46), we arrive at the rotatory-strength sum rule for natural circular dichroism,

$$\left\langle \operatorname{Im} \sigma^{\mathrm{A}}_{ab,c}(\omega) \right\rangle = 0\,. \tag{56}$$

This sum rule is well known for molecules in solution [5], as well as for oriented molecules [6]. Here, we have relied on a topological argument [23] to show that it remains valid for crystals, both insulating and conducting. Alternative discussions restricted to insulators are given in Refs. [19, 20].

The above derivation highlights the connection between the oscillator-strength and natural rotatory-strength sum rules for crystals, and the equivalence between the Kubo formulas (7) and (11) – that is, the vanishing of $\Delta_{ab}(\mathbf{q})$ – at zero and first order in $\mathbf{q}$, respectively. More generally, the expansion of Eq. (53) in powers of $\mathbf{q}$ yields at each order two optical sum rules, one of which relies on the vanishing of $\Delta_{ab}(\mathbf{q})$ at that order.

# 6 A tight-binding example

In this section, we use numerical tight-binding calculations to validate our formalism, and to illustrate the distinctive low-frequency profiles of the rotatory power in insulators and conductors.

As a simple model of a bulk crystal with nonzero $\sigma^{\mathrm{A}}_{ab,c}$, we take the tight-binding model of Ref. [58], which consists of honeycomb layers coupled by a chiral pattern of interlayer hoppings. To break time-reversal symmetry, so that $\sigma^{\mathrm{S}}_{ab,c}$ becomes nonzero as well, we add complex intralayer hoppings. The resulting model is depicted in Fig. 1(a), and its Hamiltonian reads

$$\mathcal{H} = \Delta \sum_i \xi_i c_i^\dagger c_i + i t_1 \sum_{\langle i,j \rangle} \xi_j c_i^\dagger c_j + \frac{i\lambda_1}{a} \sum_{\langle i,j \rangle} c_i^\dagger \left( \boldsymbol{\sigma} \cdot \boldsymbol{\delta}_{ij} \right) c_j + \frac{i\lambda_2}{a} \sum_{[i,j]} c_i^\dagger \left( \boldsymbol{\sigma} \cdot \mathbf{d}_{ij} \right) c_j\,. \tag{57}$$

The first term is a staggered on-site potential, with $\xi_i = \pm 1$ for the two sublattices in each layer. The second and third terms describe intralayer hoppings between nearest-neighbor sites $i$ and $j$: the second is the complex hopping responsible for breaking time reversal, and the third is a spin-orbit coupling term; therein, $\boldsymbol{\sigma}$ is the vector of Pauli matrices and $\boldsymbol{\delta}_{ij}$ is the vector taking from site $j$ to site $i$. The last term is the helical pattern of interlayer hoppings that renders the model chiral, with $\mathbf{d}_{ij}$ the vector taking from site $j$ to site $i$ in adjacent layers. We choose the distance $a$ between nearest-neighbor sites on the same layer as the unit of length, and the nearest-neighbor hopping amplitude $t_1$ as the unit of energy. For our tests, we set $c = 1$, $\Delta = 0.5$, $\lambda_1 = -0.06$, and $\lambda_2 = 0.05$.

Exploiting the translational symmetry of the crystal, we replace the site indices $\{i\}$ with $\{\mathbf{R}i\}$, where the lattice vector $\mathbf{R}$ labels the cell, and $i$ is now an intracell site index. The Hamiltonian matrix elements are denoted by $\mathcal{H}_{ij}(\mathbf{R}) = \langle \phi_{\mathbf{0}i} | \mathcal{H} | \phi_{\mathbf{R}j} \rangle$, where $\phi_{\mathbf{R}j}(\mathbf{r}) = \varphi_j(\mathbf{r} - \mathbf{R} - \boldsymbol{\tau}_j)$ is a basis orbital centered at $\mathbf{R} - \boldsymbol{\tau}_j$ [39]. The tight-binding Hamiltonian in $\mathbf{k}$ space is constructed as

$$H^{\mathbf{k}}_{ij} = \sum_{\mathbf{R}} e^{i\mathbf{k} \cdot (\mathbf{R} + \boldsymbol{\tau}_j - \boldsymbol{\tau}_i)} \mathcal{H}_{ij}(\mathbf{R})\,, \tag{58}$$

leading to the eigenvalue equation $H_{\mathbf{k}} \cdot C_{n\mathbf{k}} = \varepsilon_{n\mathbf{k}} C_{n\mathbf{k}}$.

The energy bands of the model are displayed in Fig. 1(b). There are two composite groups with two bands each, separated by a gap. We treat the lowest group as occupied, and calculate

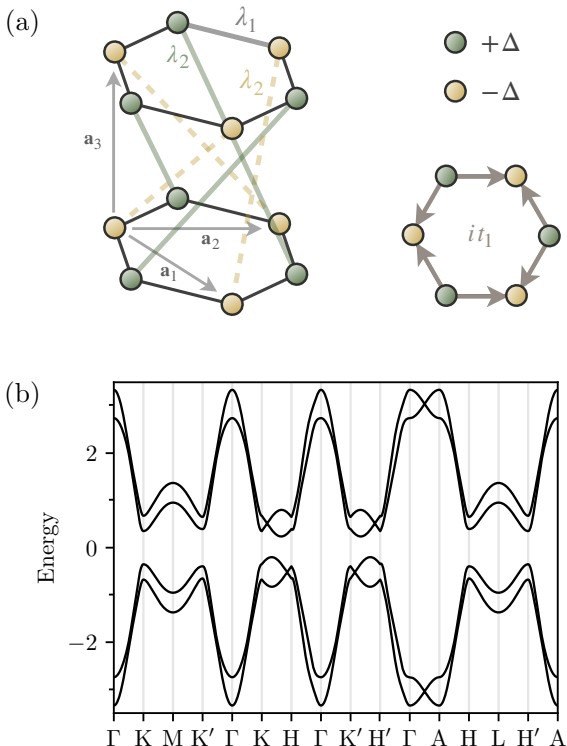

Figure 1: (a) The tight-binding model of Eq. (57). The crystallographic vectors are $\mathbf{a}_1 = (\sqrt{3}a, 0, 0)$, $\mathbf{a}_2 = (\sqrt{3}a/2, 3a/2, 0)$, and $\mathbf{a}_3 = (0, 0, c)$, with $a$ the distance between nearest-neighbor sites on the same layer, and $c$ the interlayer distance. The on-site energies and hoppings are indicated schematically. As shown on the right, the complex intralayer hoppings from $+\Delta$ sites to $-\Delta$ sites have amplitude $it_1$; the reverse hoppings (not shown) have amplitude $-it_1$. (b) Band structure of the model for the Hamiltonian parameters given in the main text.

$\sigma_{ab,c}^{\text{A}}$ and $\sigma_{ab,c}^{\text{S}}$ at zero temperature using Eqs. (26) and (31), respectively. The $A^a$ and $B^{ab}$ matrices are evaluated from Eqs. (24) and (25), with effective velocity matrix elements given by [47]

$$\mathbf{v}_{nl}(\mathbf{k}) = \frac{1}{\hbar} C_{n\mathbf{k}}^{\dagger} \cdot (\partial_{\mathbf{k}} H_{\mathbf{k}}) \cdot C_{l\mathbf{k}}. \tag{59}$$

As the system is insulating, the only nonzero contributions to $\sigma_{ab,c}$ come from the Fermi-sea terms in Eqs. (26) and (31); we will restrict our calculations to frequencies well below the threshold for interband absorption, where one can safely set $\eta = 0$ in those equations.

The magnetic point group of the model is 32. Of the four independent tensor components that are allowed by symmetry [67, 68], $\sigma_{yz,x}^{\text{A}}$, $\sigma_{xy,z}^{\text{A}}$, $\sigma_{xx,y}^{\text{S}}$ and $\sigma_{xz,y}^{\text{S}}$, only the first three are actually nonzero when the Fermi level $\varepsilon_{\text{F}}$ lies in the gap. Converged results, obtained by sampling the Brillouin zone on a uniform mesh of $50 \times 50 \times 50$ $k$ points, are shown as solid lines in Fig. 2.

For comparison, we show as filled circles in Fig. 2 the results obtained from calculations on finite crystallites. We treat them as "molecules," and evaluate $\sigma_{ab,c}^{\text{A}}$ and $\sigma_{ab,c}^{\text{S}}$ from Eq. (39) under open boundary conditions. Calculations are performed for samples with $L + 1$ unit cells in each crystallographic direction, with $L$ ranging from 4 to 12. The results are extrapolated to $L \to \infty$ by fitting them to the function

$$f(L) = f_0 + f_1/L + f_2/L^2 + f_3/L^3, \tag{60}$$

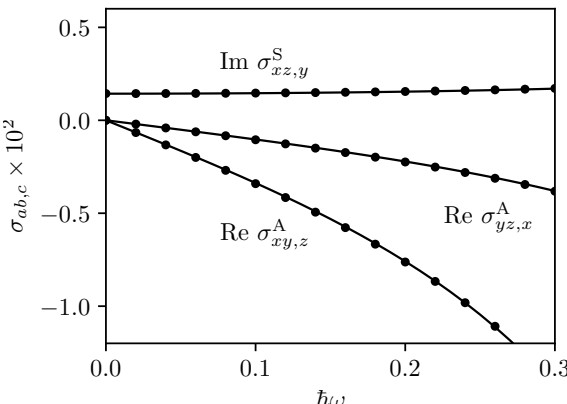

Figure 2: Numerical results for the nonzero components of $\sigma_{ab,c}$ for the model of Eq. (57) and Fig. 1, with the two lowest bands treated as occupied. The results are plotted as a function of frequency up to $\hbar\omega = 0.3$, which is well below the threshold for interband absorption ($\varepsilon_{\text{gap}} \approx 0.53$). Filled circles: extrapolation from finite-size crystallites. Solid lines: bulk crystal. The tensor $\sigma_{ab,c}$ has been divided by $e^2/\hbar$ to make it dimensionless.

where $f_0$ is the extrapolated value to be compared with the result of the bulk calculation, and $f_1/L$, $f_2/L^2$, and $f_3/L^3$ account for face, edge, and corner corrections, respectively [69]. The excellent agreement seen in Fig. 2 between the two types of calculations confirms the validity of our formalism for band insulators.

In Fig. 3, we take the $\sigma_{xy,z}^{\text{A}}$ curves from Fig. 2 and split them into origin-independent contributions. For the extrapolated crystallites (left panel), there are two types of contributions in Eq. (39): those containing $\bar{m}$ are denoted as $\bar{\text{M}}1$, and those containing $\bar{q}$ are denoted as $\bar{\text{E}}2$. For the bulk crystal (right panel), there are, in addition to $\bar{\text{M}}1$ and $\bar{\text{E}}2$ contributions from the first term in Eq. (26), band-dispersion contributions from the second and third terms, which are denoted as **v**.

A comparison between the two panels of Fig. 3 reveals that the $\bar{\text{M}}1$ and $\bar{\text{E}}2$ contributions are different for the extrapolated crystallites and for the bulk crystal, with the difference being exactly compensated by the **v** contributions that are only present in the latter. A similar situation occurs for the ground-state orbital magnetization of a crystal, whose bulk expression contains a subtle Berry-curvature term without an obvious counterpart in the molecular theory [69,70]. For a two-dimensional insulator with a single valence band, that term reads $-(e/2\hbar)\int_{\mathbf{k}} \varepsilon_{\mathbf{k}}(\partial_x \mathcal{A}_y - \partial_y \mathcal{A}_x)$ or, after integrating by parts, $(e/2)\int_{\mathbf{k}}(v_x \mathcal{A}_y - v_y \mathcal{A}_x)$. As in Fig. 3, this additional band-dispersion contribution must be included in the bulk calculation to recover the net orbital magnetization of a large flake [69,70].

To conclude, let us illustrate the different low-frequency behaviors of the rotatory power in insulating and conducting states of our model. We evaluate $\rho(\omega, \tau)$ for the bulk model from Eqs. (26) and (28), setting $\hbar/\tau = 2 \times 10^{-3}$. The frequency range is chosen as $0 \leq \hbar\omega \leq 10^{-2}$, and the calculation is carried out at zero temperature for $\varepsilon_{\text{F}} = 0.0$ (insulator) and $\varepsilon_{\text{F}} = 1.0$ (metal). In both cases, a uniform mesh of $100 \times 100 \times 100$ $k$ points is used to sample the Brillouin zone; to improve the convergence of the calculation in the metallic case, the Fermi-surface terms in Eq. (26) are evaluated as Fermi-sea integrals by performing an integration by parts.

The resulting $\rho(\omega)$ profiles, plotted in Fig. 4 as solid lines, display the behavior dictated by Eq. (29). The insulator displays a simple $\rho \propto \omega^2$ decay with a negligible influence from the scattering time $\tau$. Instead, for the metal one can distinguish two different parabolic regimes (dashed lines) delimited by $\omega\tau \sim 1$, in accordance with Eq. (30). This distinctive

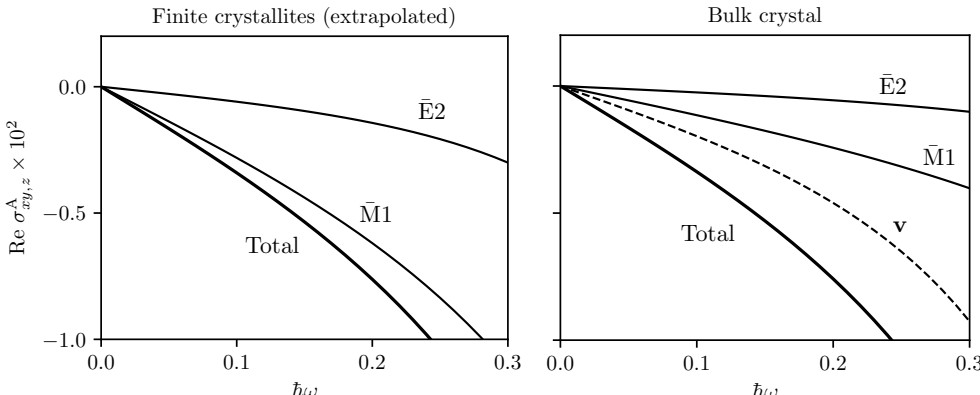

Figure 3: Decomposition of the $\sigma^{\text{A}}_{xy,z}$ curves in Fig. 2 into three types of origin-independent contributions: intrinsic magnetic dipole ($\bar{\text{M}}1$), intrisic electric quadrupole ($\bar{\text{E}}2$), and band dispersive ($\mathbf{v}$). The latter is only present in the bulk calculation on the right, and it must be included to obtain the same total result as in the extrapolated crystallite calculations on the left.

low-frequency profile of the optical rotatory dispersion in conducting crystals awaits experimental verification.

## 7  Summary and discussion

In summary, we have developed a band-theoretical description of optical spatial dispersion in insulating and conducting crystals. The novelty with respect to previous formulations resides in the fact that the current induced by the optical field is given in a physically-transparent form, as a sum of contributions that are individually origin independent, and which remain invariant under single-band gauge transformations of the Bloch eigenstates. Although we have focused on the optical conductivity $\sigma_{ab,c}(\omega)$ at first order in $\mathbf{q}$, higher-order responses can in principle be treated in a similar manner.

For a crystal composed of nonoverlapping units, our formula for $\sigma_{ab,c}(\omega)$ reduces to the standard multipole-theory expression for molecules, but with the transition moments $(d, m, q)$ of Eq. (1) replaced by their intrinsic (origin-independent) counterparts $(\bar{d}, \bar{m}, \bar{q})$ given by Eq. (2). Away from the molecular limit, $\sigma_{ab,c}(\omega)$ changes in two ways. First, the intrinsic transition moments between delocalized Bloch eigenstates are no longer given by Eq. (2); instead, one should use either the quantum-geometric expressions in Eq. (23), or their sum-over-states counterparts in Eq. (24). The second change is that $\sigma_{ab,c}(\omega)$ acquires additional band-dispersion contributions associated with electron transfer between crystal cells; this is in line with the modern theories of electric polarization and orbital magnetization in crystals [39].

There were two key aspects to our derivation. The first was the use of covariant Bloch-state derivatives to expand the optical conductivity in powers of $\mathbf{q}$; this allowed us to eliminate spurious noncovariant terms, and to isolate the physically relevant matrix elements $\bar{d}$, $\bar{m}$, and $\bar{q}$. The second was the choice of the nonsingular form of the Kubo formula in Eq. (11), rather than Eq. (7), as the starting point for the expansion in $\mathbf{q}$. An order-by-order analysis of the equivalence between the Kubo formulas (7) and (11) led us to identify a hierarchy of optical sum rules. In particular, we found that the well-known rotatory-strength sum rule from molecular physics remains valid for crystals, thanks to a topological argument involving the $\mathbf{k}$-space Berry curvature.

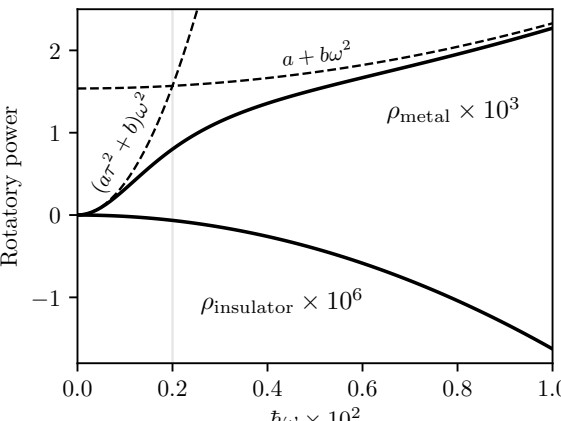

Figure 4: Low-frequency optical rotatory dispersion of the model of Eq. (57) and Fig. 1 in an insulating state ($\varepsilon_F = 0.0$), and in a metallic state ($\varepsilon_F = 1.0$). Numerical results based on Eq. (26) and Eq. (28) are shown as solid lines. The dashed lines are guides to the eye, indicating the distinct parabolic behaviors in the metallic state for $\omega\tau \ll 1$ and for $\omega\tau \gg 1$, as per Eq. (30), and the faint vertical line indicates the crossover frequency $\omega = 1/\tau$. The rotatory power has units of radians per length.

Our work opens up new prospects for realistic *ab initio* calculations of spatially-dispersive optical responses in crystals, and of static magnetoelectric and quadrupolar responses as well. An implementation based on the sum-over-states formulas for $(\bar{d}, \bar{m}, \bar{q})$ in Eq. (24) has already been carried out in a concurrent work done in coordination with the present one [71], and we also envision evaluating the **k**-derivative formulas in Eq. (23) using Wannier interpolation [72].

In closing, we mention a possible connection with the theory of the orbital Hall effect. It was recently proposed [73–75] to evaluate the orbital Hall conductivity using an orbital-current operator whose matrix elements in the Bloch-eigenstate basis are proportional to $\sum_l \left( m^{z,\mathrm{orb}}_{ml} v^a_{ln} + v^a_{ml} m^{z,\mathrm{orb}}_{ln} \right)$. Here, $\mathbf{m}^{\mathrm{orb}}_{ln}$ is the bulk generalization of Eq. (1b), given by the same expression as in Eq. (23b) for $\bar{\mathbf{m}}^{\mathrm{orb}}_{ln}$, but with the covariant derivatives therein replaced by ordinary derivatives. The two definitions are related by

$$\bar{\mathbf{m}}^{\mathrm{orb}}_{ln} = \mathbf{m}^{\mathrm{orb}}_{ln} - \frac{e}{4i}\omega_{ln}(\mathcal{A}_l + \mathcal{A}_n) \times \mathbf{A}_{ln}, \tag{61}$$

and therefore they agree for $l = n$ only. For $l \neq n$, the two terms on the right-hand side of Eq. (61) are not separately gauge covariant, and the lack of gauge covariance of $\mathbf{m}^{\mathrm{orb}}_{ln}$ makes the orbital Hall conductivity gauge dependent. This suggest that one should generally use $\bar{\mathbf{m}}^{\mathrm{orb}}_{ln}$ instead of $\mathbf{m}^{\mathrm{orb}}_{ln}$ when evaluating the orbital Hall conductivity. In other words, one is allowed to work with $\mathbf{m}^{\mathrm{orb}}_{ln}$ only in the parallel-transport gauge, where $\mathcal{A}_l = \mathcal{A}_n = \mathbf{0}$.

# Acknowledgements

This work was supported by the IKUR Strategy under the collaboration agreement between the Ikerbasque Foundation and the Material Physics Center on behalf of the Department of Education of the Basque Government, and by Grant No. PID2021-129035NB-I00 funded by MCIN/AEI/10.13039/501100011033. The authors wish to thank Cheol-Hwan Park, Stepan Tsirkin, Xiaoming Wang, Fernando de Juan, Raffaele Resta, and Massimiliano Stengel for valuable discussions and comments on the manuscript.

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
