# Peer review of "Multipole theory of optical spatial dispersion in crystals"

_SciPost Physics, doi:SciPost Phys. 14, 118 (2023)_

## Round 1 · Referee Report · Anonymous (Referee 1) · 2023-2-2

Strengths

1-A very detailed rigorous derivation of natural optical activity and of the key ingredients, the multipole transition moments.
2-Excellent, thorough overview of the existing literature and the issues. In particular, for any term, it is referred whether it appeared in previous derivation.
3-Beside the derivation, sum rules, the physical meaning of each term and its molecular limit are given.
4-The conclusions provide concrete examples of application of the formulae beyond the work. In fact, a concurrent work has implemented the formulae within an ab-initio context

Weaknesses

1- It is a long-read and not a light one, even though the authors detailed a lot, the reader needs to work a bit in order to follow the details. This is 'intrinsic' in this type of research work.

Report

This work can be used as a reference by specialists working in the field. Non-specialists which can be interest (e.g. user of codes that employ the derived expressions) can still refer to the well-written introduction and conclusions for the background, main results and significance of the work. I do not have to propose particular changes to the work, which I believe should be published on Scipost as it met these criteria:
1. It "connects the dots" on a previously-identified and long-standing research stumbling block
2. It is written clearly and precisely
3. As reported in the strengths, the introduction provides the context, the work is well detailed, the existing literature correctly referred, results are objectively summarised, conclusions are clear, linking with the rest of the work and offering perspectives of future work.

Requested changes

No changes to suggest. A minor point, I believe that the '-' at the end of the first line of (16d) should be a '+'.

  • validity: high
  • significance: good
  • originality: high
  • clarity: high
  • formatting: excellent
  • grammar: excellent

Author:  Óscar Pozo Ocaña  on 2023-02-23  [id 3397]

(in reply to Report 1 on 2023-02-02)
Category:
correction

We thank the referee for the thoughtful report. We have implemented the suggested change, by removing the '-' at the end of the first line of (16d) [now (14d)].

---

## Round 1 · Referee Report · Anonymous (Referee 3) · 2023-2-7

Strengths

1. Detailed and concise formulation.
2. The formulation is numerical friendly.

Weaknesses

Lengthy

Report

This work provides systematic and very detailed derivations of the multipole theory of spatial dispersion in crystals. It started with the expansion of the interaction Hamiltonian in the velocity form, and the multipole moments were obtained in the length form. The formulation of the individual gauge invariant components is not only numerical friendly, but also can provide much physical insights. In addition, the band dispersion term was clearly formulated which was usually neglected in the quantum chemistry community. Finally, the formulation was reduced to the molecular theory but with the origin problem eliminated, which cannot be easily solved for the quantum chemistry formulation. It is expected that the work may stimulate further ab-initio works due to the clear and easy-to-implement formulation and open the way towards light-matter phenomena of solids beyond dipole approximation. In summary, the manuscript is already in well-conditioned form. I have no comments or suggestions on any change.

Requested changes

N/A

  • validity: high
  • significance: high
  • originality: high
  • clarity: high
  • formatting: excellent
  • grammar: excellent

Author:  Óscar Pozo Ocaña  on 2023-02-23  [id 3399]

(in reply to Report 3 on 2023-02-07)
Category:
remark

We thank the Referee for the thoughtful report.

---

## Round 1 · Referee Report · Anonymous (Referee 2) · 2023-2-7

Report

This feels like a solid theoretical work, and a timely contribution to a research topic that is attracting increasing interest. The somewhat heavy algebra presented here reflects the difficulties that one has to face in dealing with spatial dispersion effects in the framework of first-principles theory; the authors do a good job at discussing the related formal subtleties in an orderly and clear way. Extensive and careful referencing to the relevant literature is provided in order to place the material in its context. Overall, I believe that this is a quality work that is likely to become a milestone in its field.

I am leaning towards recommending publication in the present form, but first I'd encourage the authors to consider the following (optional) questions.

As far as I understand, the origin dependence of the traditional quadrupolar and magnetic contributions to the gyration tensor can be traced back to the gauge freedom of electromagnetism: the magnetic moment of a current-density field J, for example, is well defined only if J(r) is purely solenoidal. Here the authors demonstrate that these contributions can be hacked in a way that they become separately origin independent.

1) Does this operation have an obvious physical interpretation (e.g., in terms of fixing the EM gauge) or should it be mostly regarded as a convenient trick to improve the numerics (preventing basis set truncation errors etc.)?

2) Related question: are the two "intrinsic" multipolar contributions that the authors propose here separately meaningful (and possibly measurable), as one would expect from gauge-invariant quantities? How about the five terms in Eq.(28)?

  • validity: -
  • significance: -
  • originality: -
  • clarity: -
  • formatting: -
  • grammar: -

Author:  Óscar Pozo Ocaña  on 2023-02-23  [id 3398]

(in reply to Report 2 on 2023-02-07)
Category:
answer to question

We thanks the Referee for the thoughtful report. In the following, we answer the questions that were raised. Whenever specific equations or references are mentioned, they refer to the resubmitted version of the manuscript.

1) We agree that the origin dependence of the traditional quadrupolar and magnetic contributions to the gyration tensor can be traced back to the gauge freedom of electromagnetism; this is clearly discussed in Sec. 3.3.1 of Ref. 9.

On the other hand, the way in which we have "hacked" those contributions was by enforcing gauge invariance in a different sense: invariance under k-dependent phase changes of the Bloch eigenstates.

We do not have a clear idea on how to relate our "hack" to electromagnetic gauge transformations, but we feel that it would be worthwhile to investigate this matter further. At present, we regard it as a convenient formulation that remains well-defined for crystals, and which may lead to improved numerics for molecules.

2) This is another excellent question, and in this case we believe we can provide a more definite answer.

It is indeed possible, on the basis of purely phenomenological considerations, to break down the optical conductivity at first order in q into separately meaningful parts, namely, into magnetoelectric and quadrupolar contributions. This was recognized in the seminal work of Ref. 3, and Ref. 4 discusses how a purely quadrupolar optical response can be isolated for some crystal classes.

More generally, magnetoelectric and quadrupolar responses can be in principle measured separately in the static limit where electric and magnetic fields become decoupled. For example, the fifth term in Eq. (26) [the old Eq. (28) mentioned by the Referee] describes, in the static limit, a "current-induced magnetization" (or "kinetic magnetoelectric effect", also known as "Edelstein effect") in gyrotropic conducting crystals.

To address this important point, and also to help readers make sense of the long equations derived in Sec. 3, we have added at the end of that section a new subsection where these matters are briefly discussed. Equation (35), which reveals the physical meaning of each separate term, is the key result of that subsection.

---

## Round 2 · Referee Report · Anonymous (Referee 1) · 2023-2-28

Report

I read the authors' resubmission and I confirm my previous positive review.

---

## Round 2 · Referee Report · Anonymous (Referee 2) · 2023-3-2

Report

The authors have provided satisfactory answers to my questions, and I am generally happy with the revision of the text. There is only one sentence that caught my attention, and should probably be fixed: at page 6 "similar comments apply to first-principles calculations using nonlocal pseudopotentials." What do the authors mean here? A theory of the current-density response in presence of nonlocal pseudos was presented in

https://journals.aps.org/prb/abstract/10.1103/PhysRevB.98.075153

This work seems to suggest that even Eq.(6) of the present manuscript should be revised in presence of nonlocal pseudos. Is this what the authors meant with their comment? It might be worth rewording it, as it sounds ambiguous in the present form. While I do recommend such a revision, I leave it to the discretion of the authors.
  • validity: -
  • significance: -
  • originality: -
  • clarity: -
  • formatting: -
  • grammar: -

Author:  Óscar Pozo Ocaña  on 2023-03-06  [id 3443]

(in reply to Report 2 on 2023-03-02)
Category:
correction

We thank the referee for raising the issue of nonlocal pseudopotentials, and for providing the very useful reference. In view of that work, we feel that this issue needs to be investigated further. For the purpose of the present work, where only tight-binding calculations are presented, the sentence about nonlocal pseudopotentials can be safely removed. We have done so in the resubmitted manuscript, and we now mention above Eq. (6) that a local external potential is assumed.

---

## Round 2 · List of Changes

• Added a new section (Sec. 3.6), and made corresponding changes in the Introduction and Summary sections.

  • The titles of Secs. 3, 3.1, 3.2, 3.3, 3.5.1, and 3.5.2 have been reworded for clarity.

  • Added, at the end of Sec. 2 and beginning of Sec. 3.5, a discussion of the restrictions imposed on the spatially-dispersive conductivity by inversion (P), time reversal (T), and by the combined PT symmetry.

  • Some reorganization around Eqs.(11,12), which made the text shorter.

  • Factorized an expression in Eq. (14d).

  • Dropped an incorrect sentence below Eq. (27).

  • Dropped a sentence below Eq. (32).

  • A (formally vanishing) term was added to Eq. (47) for clarity.

  • Corrected a typo in the second term of Eq.(57).

  • Made some changes in Fig. 1(a) (right side), and added an explanatory sentence in the caption.

  • Dropped a sentence below Eq. (58).

  • Dropped from Sec. 6 a speculative discussion about future prospect for numerical calculations on conducting crystallites.

  • Fixed the labels of the dashed curves in Fig. 4 (they were swapped).

  • Shortened the "Summary and discussion" section.

  • Added several new references, and removed a few.

  • Fixed several typos throughout the manuscript, and made changes in wording at various places to improve the clarity of the text.

---

## Round 3 · List of Changes

- According to the discussion with referee 2, a sentence was added above Eq. (6) limiting our analysis to local external potentials.

- A typo was corrected in Eq.(33c), namely i/4 --> 1/(4i).

- The words "constitutive relation" were added above Eq. (35).

- Below Eq. (35), "[...] _uniform_ electric field".

- In the following paragraph, "in insulators" was removed (redundant).

- At the bottom of p. 2, in the paragraph starting with "By comparison [...]", "optical rotation" became "optical activity", and a new
reference (Ref. 37) was added (a very recent review about chiral thin films).

---

## Editorial Decision

published